# Probing Biological and Artificial Neural Networks with Task-dependent Neural Manifolds

Michael Kuoch[1,2]*, Chi-Ning Chou[2,3]*, Nikhil Parthasarathy[2,3], Joel Dapello[1,4],
James J. DiCarlo[1], Haim Sompolinsky[4,5], SueYeon Chung[2,3]
[1]Masssachusetts Institute of Technology, [2]Flatiron Institute, Simons Foundation,
[3]New York University, [4]Harvard University, [5]Hebrew University of Jerusalem
mikuoch@mit.edu, {cchou, schung}@flatironinstitute.org

Recently, growth in our understanding of the computations performed in both biological and artificial neural networks has largely been driven by either low-level mechanistic studies or global normative approaches. However, concrete methodologies for bridging the gap between these levels of abstraction remain elusive. In this work, we investigate the internal mechanisms of neural networks through the lens of neural population geometry, aiming to provide understanding at an intermediate level of abstraction, as a way to bridge that gap. Utilizing manifold capacity theory (MCT) from statistical physics and manifold alignment analysis (MAA) from high-dimensional statistics, we probe the underlying organization of task-dependent manifolds in deep neural networks and macaque neural recordings. Specifically, we quantitatively characterize how different learning objectives lead to differences in the organizational strategies of these models and demonstrate how these geometric analyses are connected to the decodability of task-relevant information. These analyses present a strong direction for bridging mechanistic and normative theories in neural networks through neural population geometry, potentially opening up many future research avenues in both machine learning and neuroscience.

## 1. Introduction

Unsupervised learning, specifically unsupervised deep neural networks (DNNs), have become increasingly prominent in the landscape of modern machine learning due to their ability to learn useful statistics and representations unlabeled data. They provide advantages over classical supervised DNNs in terms of cost, flexibility, and performances in various applications including image recognition [1], natural language processing [2], speech processing [3], and beyond. Despite this popularity, we still lack an intuitive and mechanistic understanding of how these unsupervised models differ from their supervised counterparts. Traditional performance metrics are limited because they focus only on end performance, without opening the "black box" of DNNs.

Meanwhile, unsupervised neural networks have gained popularity in the neuroscience community as promising models of the brain. Like biological neural networks, unsupervised DNNs can learn useful information about inputs without relying on large amounts of labeled data. Furthermore, previous work has found that unsupervised DNNs generate representations similar to the brain in terms of prediction accuracy of neural data [4], and similar to those using supervised training [5]. However, these metrics are limited because they provide similar results for many different types of models, and they can not explain why the certain models are more similar and what this similarity means mechanistically in terms of task-performance.

In this work, we investigate these questions via a framework based on neural population geometry. Our framework offers insights into why unsupervised approaches excel at certain tasks and could lead to potential strategies for designing improved learning algorithms. Indeed, recent work has demonstrated how such an intermediate understanding can facilitate robustness in self-supervised

---

*Equal contribution.

First Conference on Parsimony and Learning (CPAL 2024).

learning [6]. In parallel, a better understanding of the internal mechanisms of unsupervised DNNs could potentially illuminate the underlying learning principles adopted by the brain [7], as well as provide new ways to compare DNNs to the brain.

**Neural population geometry and organization hypotheses.** Neural population geometry [8] refers to the study of the connection between the high-dimensional geometry of neural representations (i.e., the collections of neural activities) and the underlying computations (e.g., task performance) [9, 10]. Through intuitive geometric and statistical quantitative measures (e.g., manifold classification capacity, participation ratio, intrinsic dimension, etc.), it naturally serves as an intermediate language to bridge high-level computational principles and detailed neural mechanisms.

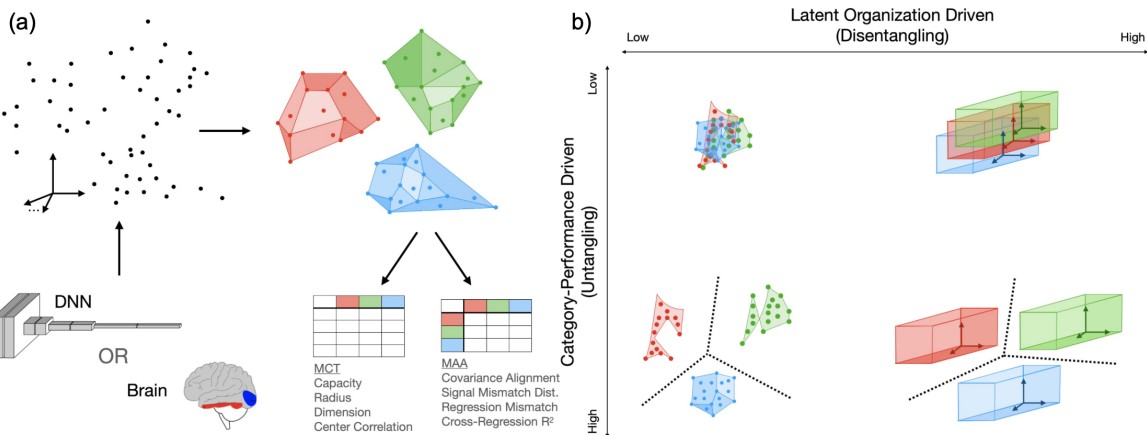

Figure 1: (a) We measure geometrical properties of task-dependent manifolds from DNNs and macaque brain neural recordings. (b) Potential strategies (Category-Performance Driven Hypothesis and Latent Organization Driven Hypothesis) used to organize representations in neural networks.

We examine the organizations of neural representations to investigate the differences between distinct learning paradigms in artificial and biological neural networks. Concretely, we study the following two organization hypotheses in this paper: (i) the Category-Performance Driven Hypothesis and (ii) the Latent Organization Driven Hypothesis. As illustrated in Figure 1(b), the former hypothesizes that neural representations are organized in a way that makes the object manifolds easily separable (a.k.a., untangling), while the latter hypothesizes that neural representations are organized according to latent information in the stimuli (a.k.a., disentangling). We utilize several geometric and quantitative tools to pinpoint the extent that each of these hypothesis manifest in supervised and unsupervised learning algorithms as well as in recordings from macaque visual cortex.

**An overview on our methods and results.** Task-dependent manifolds refer to neural manifolds that are associated with a certain computational task. In this work, we are interested in two types of task-dependent manifolds: (i) *the object manifold*, which corresponds to the neural representations of a stimuli in a classification task; (ii) *the latent variation manifold*, which corresponds to the neural representations labeled with a latent feature in a regression task. We utilize Manifold Capacity Theory (MCT) [11] and Manifold Alignment Analysis (MAA) to analyze the task-dependent manifolds of supervised and unsupervised DNNs, as well as compare these representational properties to manifolds from macaque visual cortex. While MCT provides geometric and quantitative measures (e.g., manifold capacity, radius, and dimension) on object manifolds to understand the linear classification performance of a neural network, MAA further investigates the organizations of multiple latent variation manifolds by considering the manifold orientation and alignment. Armed with these new geometrical viewpoints, we present the following findings:

- (**The geometry of object neural manifolds**) The representations of supervised and unsupervised DNNs differ by their size, with supervised models achieving higher class manifold capacity by shrinking their class manifolds to a greater extent than unsupervised models.

- (**The alignment across object manifolds**) The object manifolds of unsupervised DNNs are more aligned in the representational space then the object manifolds of supervised DNNs.
- (**Decodability of task-related information from latent variation manifolds**) Stronger manifold alignment is associated with lower regression error, suggesting a potential advantage of learning more aligned representations.

These findings together suggest that the unsupervised models and supervised DNNs differ in their organizational strategies. Supervised DNNs have representation that are more specialized for classification, displaying a higher degree of category-performance driven organization. On the other hand, Unsupervised DNNs tend to show greater latent organization driven representations that can that retain more general information about input stimuli.

## 2. Methods

### 2.1. Geometric and quantitative tools

We utilize quantitative measures from manifold capacity theory (MCT) and manifold alignment analysis (MAA) as an intermediate layer of abstraction that connects the underlying computation performed by neural networks to the geometry of neural representations. By examining these macroscopic observables of the object manifolds, we quantitatively compare different learning algorithms and investigate the organization hypotheses.

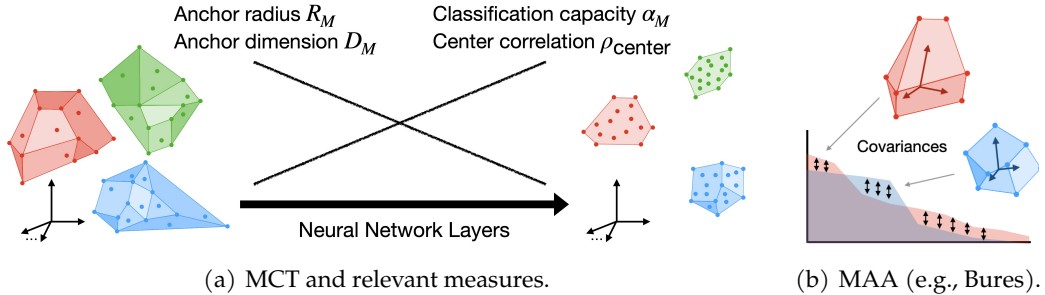

(a) MCT and relevant measures.  (b) MAA (e.g., Bures).

Figure 2: (a) In MCT, we expect the radius and dimension of manifolds will decrease across neural network layers. Meanwhile, we expect the classification capacity and the center correlation to increase. This figure is an illustration of the intuitive picture of how the organization changes. (b) The covariance alignment distance from Bures metrics is in analogous to the Wasserstein distance, which intuitively captures the minimum cost to turn one distribution into another.

**Manifold capacity theory (MCT)** [11] uses tools and concepts from statistical physics to quantify the linear classifiability of object manifolds. It has previously been used in the study of biological neural networks [12, 13], deep learning models [8, 10, 14], and self-supervised learning [6]. The resulting measure of manifold classification capacity, denoted $\alpha_M$, takes value between 0 and 2 where the higher the capacity, the more efficient the neural network is representing the object. The derivation using replica method from theoretical physics further reveals that $\alpha_M$ is linked to two geometric quantities of the manifold, namely the *anchor radius* $R_M$ and the *anchor dimension* $D_M$. Specifically, $\alpha_M$ can be approximated by $(1 + R_M^{-2})/D_M$ [11]. That is, the smaller the radius and the dimension, the more efficient the neural representations are. We also adopt from MCT the center correlation $\rho_{center}$, which measures how correlated the center locations of each object manifold are. Concretely, high $\rho_{center}$ would suggest that the manifolds are clustering in the neural activity space. The intuitive picture of the quantitative measures discussed above are summarized in Figure 2(a). We use **MCT to probe how much neural networks utilize the category-performance driven organizational hypothesis**, i.e., higher classification capacity corresponds to a higher degree of category-performance driven organization.

**Manifold alignment analysis (MAA).** MCT provides us with information on manifold capacity, dimension, and radius from the perspective of a linear separability of the object manifolds. However,

these properties might not capture the underlying latent organizations of manifolds. As such, we propose additional metrics to complement the geometrical picture revealed by MCT. Specifically, we measure how aligned manifolds are in the representational space to help us identify the extent of latent organization in the representations (Figure 1b, bottom row). We use **MAA to probe how much neural networks utilize the latent-organization driven organizational hypothesis**, i.e., higher degree of alignment corresponds to a higher degree of latent-organization, as latent information becomes more structured in the manifold representations. In the following we introduce two of the main metrics used in MAA and leave the rest in the Appendices.

*Covariance alignment distance from Bures metric.* One approach to measuring manifold alignment is to model each object manifold as a Gaussian distribution and measure the empirical covariance of the object representations. Under this formulation, the empirical covariance describes the orientation of the object manifold in the representational space., allowing us to use the distance between object manifolds' covairinces as a proxy for their alignment. Concretely, we use a variant of the covariance term of the Bures metric to measure differences between object covariance distances. To measure manifold alignment, we specifically compute the Bures covariance distance on the trace-normalized object covariances to remove confounding factors (such as covariance scale) that could influence the Bures metric. See Appendix B.2 for an in-depth discussion on the covariance alignment distance and see Figure 2(b) for a pictorial illustration.

*Signal mismatch distance from linear regression.* We introduce the signal mismatch distance, a geometric measure that captures the regression performance of manifold alignment. Formally, we model a manifold $M$ as $\{(x^M(\theta, \psi), \theta)\}$ where $x^M(\theta, \psi)$ is a neural representation parameterized by feature value $\theta$ and in-manifold variability $\psi$. The least squares error of linearly regressing on $M$ (without the biased term, or equivalently, being mean centered) is

$$LSE^M = \langle \theta^2 \rangle_M - (\bar{x}^M)^\top (C^M)^{-1} (\bar{x}^M)$$

where $\bar{x}^M = \langle [\theta \cdot x^M(\theta, \psi)] \rangle_M$ and $C^M = \langle x^M(\theta, \psi)(x^M(\theta, \psi))^\top \rangle_M$ and $\langle \cdot \rangle_M$ refers to taking average over the manifold $M$. Here we are interested in two manifold $A$ and $B$ as well as their union $A \cup B$. Finally, consider the difference between the least square error of the union of the two manifolds and the average of the least square error of the two individual manifolds, we have

$$\left( \frac{LSE^A + LSE^B}{2} \right) - LSE^{A \cup B} = \frac{1}{2}(\bar{x}^A - \bar{x}^B)^\top (C^A + C^B)^{-1} (\bar{x}^A - \bar{x}^B) \tag{1}$$

$$+ \frac{1}{2}(\bar{x}^A)^\top [(C^A)^{-1} - 2(C^A + C^B)^{-1}](\bar{x}^A) \tag{2}$$

$$+ \frac{1}{2}(\bar{x}^B)^\top [(C^B)^{-1} - 2(C^A + C^B)^{-1}](\bar{x}^B). \tag{3}$$

The above difference of least square errors captures the amount of signal alignment of the two manifolds. Meanwhile, under mild statistical assumptions, Equation 2 and Equation 3 are independent to the signal correlations between the two manifolds. Namely, the information about signal alignment of the two manifolds is fully contained in Equation 1. As this term naturally looks like a geometric distance, we define the (normalized) signal mismatch distance between manifold $A$ and $B$ as

$$d_{\text{signal mismatch}}(A, B) = \frac{(\bar{x}^A - \bar{x}^B)^\top (C^A + C^B)^{-1} (\bar{x}^A - \bar{x}^B)}{(\bar{x}^A)^\top (C^A + C^B)^{-1} (\bar{x}^A) + (\bar{x}^B)^\top (C^A + C^B)^{-1} (\bar{x}^B)}.$$

Signal mismatch distance is designed for probing local manifold properties generated by small variations of latent parameters around each exemplar's representation, so that irrelevant large-scale nuisance variable would not dominate the signal. See Appendix B.3 for more discussions.

## 2.2. Models and datasets.

In this paper, we utilize a dataset consisting of images of three-dimensional objects overlayed on top of a background image ([15], [16]). There are 64 different objects that can be grouped into 8 superclasses. In each of the images, the object is associated with 6 viewing parameters: size, position (x and y), and rotation (x, y, and z). Examples images are shown in Figure 3.

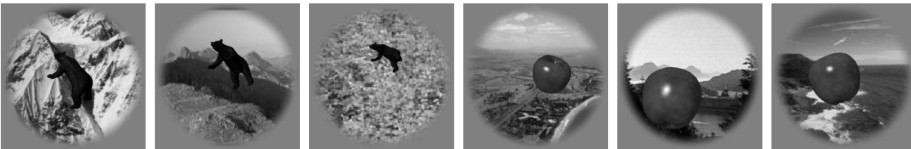

Figure 3: Examples images from the "bear" and "apple" object categories. As in [15], [16], each image is composed of a 3D object overlayed on a background image. The 3D object has six variable parameters: size, position (x and y), and rotation (x, y, and z).

We use this dataset to investigate the representations of a variety DNNs trained with different algorithms. To control for model architecture, each of the DNNs we investigated used a ResNet-50 architecture [17], eliminating variation that would arise due to architecture differences. We explored the following learning algorithms: Supervised, Supervised with random erase [18], Supervised with random erase + random augment, Barlow Twins [19], DeepClusterV2 [20][21], SWAV [21], SimCLR [22], PIRL [23], BYOL [24], VICreg [25]. Each of these models were trained on ImageNet images [26]. For the figures in this paper, we aggregate the results from supervised and unsupervised models. See Appendix B.5 for information on these models' performance and sources.

**Recordings from macaque visual cortex.** We use macaque visual cortex recordings from V4 and IT to this same set of images, using micro-electrode arrays with 96 electrodes per array being surgically implanted into the brains of macaque monkeys. See [27] for more details on visual cortex recordings.

## 3. Results

### 3.1. Manifold capacity theory and the geometry of task-dependent manifolds

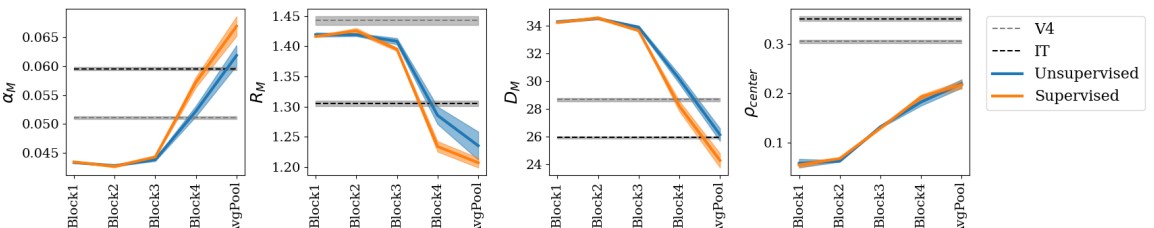

Figure 4: For each DNN, we extracted hidden representations of each image from the output of the four ResNet blocks and the terminal average pool layer. Using these representations, we computed the manifold capacity, radius, dimension, and center correlations across the layers of each DNN with MCT. We also analyzed on macaque brain recordings (dashed lines, error bars from bootstrap).

**Supervised models achieve higher classification capacity by shrinking their neural manifolds.** Our first step in investigating the organizational structure of object manifolds in deep neural networks was to probe the neural population geometry of object manifolds with MCT. As in Section 2.1, we expect the neural manifolds to "shrink", and this intuition is quantitatively characterized by the results of measurements from MCT as shown in Figure 4, aggregated by model type.

MCT reveals differences in the representations learned by supervised models and unsupervised models. Specifically, in the later layers of the DNNs we consistently see that the objects manifolds of unsupervised models are larger in both radius and dimension, and exhibit a lower manifold capacity than the supervised models. Interestingly, the center correlation is similar between both types of models across the network layers.

The trends in manifold capacity, radius, and dimension together indicate that supervised and unsupervised models utilize different strategies to organize their representations. Supervised models learn representations with high capacity by shrinking the object manifolds in their representational space, while unsupervised models learn object manifold representations that are larger and less

compressed. With regards to the organizational hypotheses established in the introduction, these results suggest that supervised DNNs exhibit a higher degree of category-performance driven organization than unsupervised models do. These differences raise interesting questions. What are the advantages and disadvantages of each organizational structure? Are supervised models learning representations that are "overspecialized" to object classification?

**Trends across macaque ventral visual stream match trends across model layers.** We repeated the MCT analysis described above for V4 and IT macaque neural recordings from to the same sets of object images (Figure 4). We see that the trend in MCT metrics across the visual cortex matches the trends we see across DNN layers (manifold capacity and center correlation increasing, manifold radius and dimension decreasing). This suggests that the high level organizational strategy of DNNs and the brains may be similar. Furthermore, we see that the terminal values of capacity, radius, and dimension in the visual cortex (IT) are closer to the terminal values of the unsupervised DNNs compared to supervised DNNs (average pool layer). This suggests that the brain, like unsupervised models, does not compress object manifolds to the extent that supervised models do.

## 3.2. Manifold alignment analysis and the geometry of task-dependent manifolds

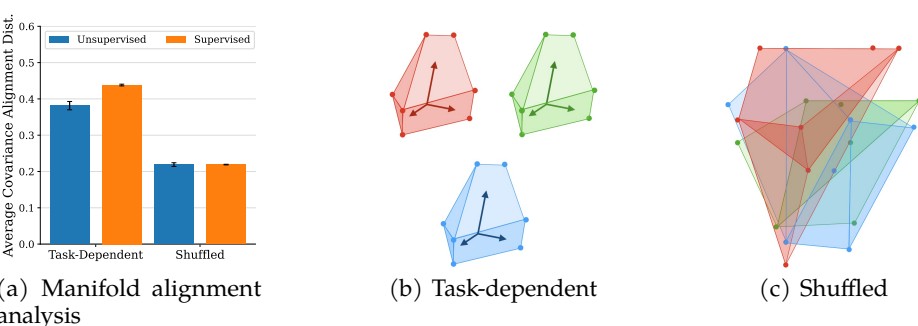

(a) Manifold alignment analysis      (b) Task-dependent      (c) Shuffled

Figure 5: (a) Unsupervised DNNs display greater manifold alignment than supervised DNNs at the terminal average pool layer. For each model, we averaged the Bures covariance distance between the 8 super categories in our dataset to quantify how aligned the object manifolds were in each model. We also repeat the experiment, but with shuffled manifolds (random partitioning). Measurements on shuffled manifolds are unable to distinguish between unsupervised and supervised models. (b-c) Task-dependent manifolds can capture meaningful structure in the representational space that is lost in shuffled (random) manifolds without task-dependence.

**The representations in unsupervised models are more driven by latent organization.** The fact that unsupervised models have larger object manifolds in terms of radius and dimension raised the interesting question of how these models can achieve good classification performance despite their larger manifolds. We realize that one way this could be archived is through aligning the manifolds in order to use the representational space more efficiently, as shown by Wakhloo et al. [28]. Thus, we decide to use MAA to investigate how aligned the object manifolds were in the representational space in order to put the MCT measurements into context. The results are shown in Figure 5(a), aggregated by model type (supervised and unsupervised). In Figure 5(b), we see results from the same experiment, but with examples randomly shuffled between manifolds. In this scenario, we are unable to see significant differences between supervised and unsupervised DNNs, demonstrating the importance of task-relevant partitions when studying these geometrical properties.

The covariance distance and signal mismatch distance measurements indicate that the representations of unsupervised models are more aligned than those of supervised models. With regards to the organizational hypotheses established in the introduction, these results suggest that unsupervised DNNs exhibit a higher degree of latent organization driven structure than supervised models do. Could these more structured representations provide advantages?

### 3.3. Relative manifold position and alignment analysis

Inspired by the discoveries in Section 3.1 , we next explore quantification of the similarity between neural activity in higher-level areas of macaque visual cortex (V4 and IT) and unsupervised vs. supervised DNNs by computing RSA-style comparisons. However, instead of using pairwise distances between individual exemplar responses, we use pairwise distances computed based on task-dependent elements (category manifold centers or category manifold covariances). We find that the geometrical properties of macaque neural recordings are consistently more similar to unsupervised DNNs than supervised DNNs. However, note that these results don't show that the brain utilizes unsupervised learning objectives. Rather, these experiments show that the organizations of representations in unsupervised DNNs more resemble those in the brain by these metrics.

**Unsupervised models position object manifolds more similarly to macaque visual cortex.** We begin by investigating the similarity between the relative positions of the object manifolds in the macaque visual cortex and in the DNNs. For the neural responses in visual cortex and each of the DNNs, we compute the centers of each of the object manifolds. Using these center locations, we compute a center correlation matrix. We correlate the off-diagonal entries of the matrices for the biological areas with those of the DNNs to measure similarity in relative object manifold positions. The results displayed in Figure 6(a) show that the representations of unsupervised DNNs are more similar to the macaque visual cortex in terms of object manifold positions. Furthermore, the similarity between the IT and DNN representations increases across the layers of the DNNs.

**Unsupervised models orient category manifolds more similarly to macaque visual cortex.** We next use a similar approach to explore a different attribute of the object manifolds: their orientation. Specifically, we compute the pairwise alignment between object manifolds (as described above) for the visual cortex and each of the DNNs to measure the orientations of the manifolds relative to each other. We then correlate the off-diagonal entries of these matrices to measure similarity in pairwise manifold alignment between the visual cortex recordings and DNNs. The results are displayed in Figure 6(b). Again, we find that the representations in the macaque visual cortex are more similar to unsupervised DNNs. These results can be interpreted as follows: two manifolds that are more aligned in the macaque visual cortex are also more aligned in the unsupervised DNNs.

**RSA is unable to distinguish different types of models.** Previously, RSA has been employed as a method to compare the similarity of DNN representations and neural recordings. In Figure 6(c), we demonstrate that merely performing RSA on pairwise distances, without any task-dependence, results in similar neural similarity scores for both supervised and unsupervised DNNs. This is unlike the outcomes observed with our proposed methods. The results underscore the importance of studying task-relevant manifolds, as opposed to responses to individual stimuli.

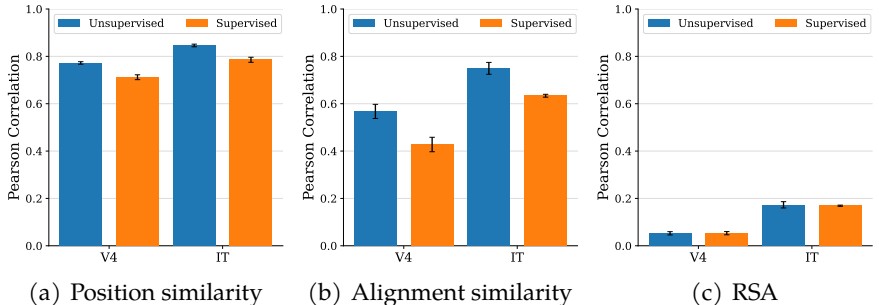

Figure 6: (a) The relative positions of neural object manifolds are more similar to unsupervised DNNs than supervised DNNs. Higher correlation means greater similarity to macaque neural recordings. (b) The relative orientations of neural object manifolds are more similar to unsupervised DNNs than supervised DNNs. (c) RSA on stimuli pairwise-distances (no task-dependence) is unable to separate the supervised and unsupervised DNN types.

### 3.4. Manifold alignment and regression performance

To investigate potential benefits of more structured representations, we next seek to examine the relationship between object manifold alignment and regression performance. To achieve this, we generate object images that varied only in one of the six viewing parameters (described in Section 2.2) to eliminate any confounding variance, and yielded latent variation manifolds (around each exemplars). Then, we utilize several quantitative measures to perform a correlation analysis to probe the potential connection between computation and geometry in the context of regression.

For each DNN, we calculate four quantities for each pair of object manifolds (average pool layer): (i) manifold covariance distance, which measures the degree of alignment of the covariance structures of the two manifolds (Section 2.1); (ii) cross-regression performance, defined as the $R^2$ achieved from training a linear regression model on one latent variation manifold around an exemplar and evaluating on another; (iii) regression performance mismatch, defined as the difference between the least squares error of the union of the two manifolds and the average of the least squares error of each manifold; and (iv) signal mismatch distance, a new distance measure we propose to quantify the regression performance (Section 2.1). While (i, ii) capture the geometry and regression performance of the manifolds respectively, (iii) and (iv) serve as quantitative measures bridging these two aspects.

We hypothesize that the more geometrically similar the two manifolds are, the better the regression performance will be. This high-level hypothesis leads to four quantitative predictions based on the aforementioned measures, as summarized in Figure 7. We found affirmative results for three out of four predictions while the failure of covariance distance positively correlating with regression performance mismatch suggests that the signal mismatch distance unveils additional computational properties of the manifolds. These findings advocate that manifold geometry may be associated with regression performance and warrant future studies.

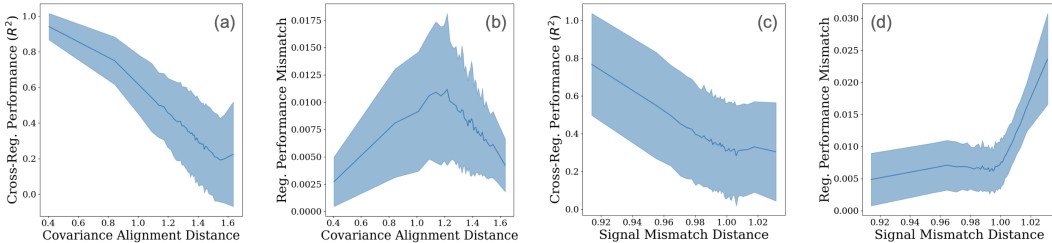

Figure 7: We had predicted that both the manifold covariance distance and the signal mismatch distance would negatively correlate with the cross-manifold regression performance and positively correlate with the regression performance mismatch. It turns out that, except for the manifold covariance distance, which does not positively correlate with the regression performance mismatch (i.e., (b)), all the other predictions (i.e., (a), (c), and (d)) are supported by the results from analyzing the terminal average pool output of each DNN. We interpret the failure of prediction (b) as suggesting the signal mismatch distance could provide extra computational information.

## 4. Related Work

**Geometric analysis in neural networks.** Previous work has utilized geometrical properties to investigate the internal mechanisms of DNNs and explain computational performance. Ansuini et al. [9] show that lower intrinsic dimension of object manifolds is associated with increased generalization performance. Cohen et al. [10] use MCT to study the way neural geometry and classification capacity change across DNN layers, and how different layers of the network can influence these properties. Yerxa et. al [6] show that training DNNs to optimize these geometrical properties can result in good classification performance. These results demonstrate the power of using geometrical tools to explain computational properties of neural networks.

**Comparing biological and artificial neural networks.** Another area of previous work has been in comparing DNNs to neural recordings to uncover which learning algorithms are most like the brain. Brain-Score [29] quantifies similarity to the brain based on PLS regression scores. DNNs are scored by their ability to predict biological neural responses to the same stimuli. Previous work [4] has used these metrics to compare DNNs to brain recordings, but these metrics are unable to provide

information about the organization of representations, which we believe to be crucial to understanding underlying mechanisms. Another method that has been used to compare computational models and biological neural responses is representational similarity analysis, RSA [30], which compares representations by computing correlations between "dissimilarity" matrices. However, recent studies have shown that these types of similarity measures have potential issues [31]. Thus, further theoretical justifications are needed to show how these metrics could bridge the representations to computational properties of interest, e.g., classification and regression performance.

**Training objectives.** Previous works have examined why certain training objectives yield better performance or generalizability, using both theoretical and empirical approaches [32–37]. Kornblith et al. [34] investigated the connection between loss functions, regularizers, and test accuracy in image classification tasks, observing that better class separability is associated with representations that are less transferable. While these studies focus more on end-performance and/or geometry in parameter space, our work complements the landscape through a geometric analysis of object representations.

**Disentangled representations.** Disentanglement, sometimes known as factorization, is a central theme in the study of neural representations. There have been efforts from both machine learning [38–40] and neuroscience perspectives [41, 42] to understand and quantify the role of disentangled (or factorized) representations. For example, $\beta$-VAEs [38] and their variants utilize information theory to encourage factorized latent distributions. Eastwood et al. [39] propose a quantitative framework for gauging the degree of disentanglement when the ground-truth latent structure is accessible. On the biological front, Whittington et al. [41] explore disentanglement through biological and normative constraints. As discussed by Locatello [43], the study of disentanglement requires insights into the inductive biases of the data, tasks, and models. While this work does not directly tackle the problem of disentanglement, we believe geometric approaches would lead to insightful understandings.

## 5. Conclusion and Discussion

In this study, we utilize Manifold Capacity Theory (MCT) and Manifold Alignment Analysis (MAA) to explore the neural population geometry of artificial and biological neural networks. Unlike traditional comparisons of different DNNs, which primarily focus on end performance rather than internal mechanisms, we demonstrate that geometric analyses at the representation level can reveal differences in the organizational strategies of DNNs with distinct objectives (supervised and unsupervised DNNs). Our findings indicate that these geometric properties are associated with computational properties of interest, such as regression performance.

While we utilize several quantitative tools to probe the organizational strategies of neural networks, some of the underlying mechanisms remain open. MCT currently focuses on the geometric properties of a single manifold and hence we develop MAA to complement the aspect of the relations across manifolds. Unlike MCT, which directly connects general geometrical properties to the efficiency of downstream computations, MAA addresses only a fraction of the picture. Nevertheless, these limitations open up numerous research questions for future work as described below.

On the theoretical side, a natural follow-up direction is to further develop rigorous connections between geometry and regression performance. In this paper, we formally motivate alignment measurements in local settings using signal mismatch distance, and we observe a striking association between alignment and regression performance in real data. A rigorous theory applicable to global manifolds linking geometry to regression performance would open many new doors for innovation in model development and understanding, just as MCT did with classification [6]. On the machine learning side, we show manifold alignment is associated with regression performance; are there other benefits? As mentioned before, manifold alignment has been connected with task-relevant properties such as capacity [28]. Is it possible that increased manifold alignment can explain other phenomena of unsupervised DNNs, such as improved performance in certain areas and the effectiveness of pretraining? A deeper understanding of how these geometrical properties influence DNN performance would be a significant asset for understanding why DNNs work. On the biological side, these findings also present new avenues for the neuroscience community to study the differences between how DNNs and the brain learn through geometrical perspectives. For exmaple, can we extend these metrics to reveal similarities and differences between different DNNs and the brain?

# Acknowledgments

We thank Uri Cohen for helpful discussions. This work was funded by the Center for Computational Neuroscience at the Flatiron Institute of the Simons Foundation and the Klingenstein-Simons Award to S.C. H.S. is supported by the Gatsby Charitable Foundation, the Swartz Foundation, and ONR grant No.N0014- 23-1-2051. All experiments were performed on the Flatiron Institute high-performance computing cluster, and the MIT OpenMind computing cluster.

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

# A. The Quantitative Measures Used in this Paper

|  | Quantitative Measures | Properties | Relevant Sections |
|---|---|---|---|
| MCT | Classification capacity $\alpha_M$ | Classification | Section 2.1, 3.1 |
|  | Anchor radius $R_M$ | Geometry + Classification | Section 2.1, 3.1 |
|  | Anchor dimension $D_M$ | Geometry + Classification | Section 2.1, 3.1 |
|  | Center correlation $\rho_{center}$ | Geometry | Section 2.1, 3.1 |
| MAA | Covariance alignment distance | Geometry + Regression* | Section 2.1, 3.2, 3.4 |
|  | PCA distance | Geometry + Regression* | Section 2.1, 3.2, 3.4 |
|  | Signal mismatch distance | Geometry + Regression | Section 2.1, 3.2, 3.4 |
|  | Relative geometry analysis | Geometry | Section 3.3 |

Table 1: A summary of the quantitative measures used in this paper. "Regression*" refers to the measure being correlated with regression performance empirically.

# B. Technical Details

## B.1. MCT

Let $M$ be an object manifold. $M$ could be empirically estimated from a neural network or a purely analytical model. The classification capacity of $M$ is defined as follows. Let $N$ be the number of neurons. For each positive integer $P$, we randomly pick the centers and coordinates for $P$ independent copies of manifold $M$ in $\mathbb{R}^N$, denoted as $M^\mu$ for $\mu = 1, 2, \ldots, P$. Next, we randomly assign $\pm 1$ label to each copy independently and denote it as $y^\mu$. Intuitively, the higher the $P$ is, the less easy to have a linear classifier to separate all these manifolds according to their labels. Concretely, the probability (over the randomness of picking centers, coordinates, and labels for each manifold) of separability has a sharp phase transition with respect to $P$ and the classification capacity is hence defined as the ration $\alpha_M = P/N$ where $P$ is the phase transition point.

In [11], Chung, Lee, and Sompolinsky used techniques from replica theory to derive an analytical expression for the capacity $\alpha_M$ in terms of the geometric properties of the manifold $M$. In particular, they showed that $\alpha_M \approx (1 + R_M^{-2})/D_M$ where $R_M$ and $D_M$ are anchor radius and anchor dimension of $M$. That is, the classification capacity and these quantities from anchor geometry serve as a bridge between the computational and representational aspect of neural manifolds. See [11] for more details.

## B.2. Covariance alignment distance as a measure of alignment

We propose measuring the alignment of covariances using a modified version of the traditional Bures metric [44], which has been shown to be a robust way to compare positive-semidefinite matrices. More importantly, it has been shown that the Bures metric can be used to compare the geometry of covariances of neural populations [45].

In it's general form, the Bures metric between two covariances $A$ and $B$ can be defined as

$$d(A, B) = Bures(A, B) = Procrustes(A^{1/2}, B^{1/2}) = \sqrt{Tr(A) + Tr(B) - 2\|A^{1/2}B^{1/2}\|_*} \quad (4)$$

where $\|.\|_*$ denotes the nuclear norm.

This metric; however, generally compares the shapes of two covariances according to a combination of their alignment in orientation (shape) *and* magnitude of variability (size). For the purposes of this work, we specifically want to understand the component of covariance similarity due to shape alignment, irrespective of size. This is because individual object manifolds may represent latent factors with different degrees of variability, but we would like to understand if in a high-dimensional space, two representations are projecting latent factors into aligned axes of variability irrespective of the size. There are many ways to normalize by magnitude of individual covariances to remove

the "size" component. Specifically, we propose evaluating the Bures metric over trace-normalized covariances and show that this distance is scale-invariant as desired:

$$d'(A, B) = d\left(\frac{A}{Tr(A)}, \frac{B}{Tr(B)}\right)$$

$$= \sqrt{2 - \frac{2\|A^{1/2}B^{1/2}\|_*}{(Tr(A^{1/2})\,Tr(B^{1/2}))}} \tag{5}$$

**Proposition 1.** *For any two positive definite matrices $A$ and $B$: $d'(sA, B) = d'(A, B)$ for an arbitrary scalar $s$.*

.

*Proof.*

$$d'(sA, B) = \sqrt{2 - \frac{2\|(sA)^{1/2}B^{1/2}\|_*}{(Tr(sA)\,Tr(B))}}$$

$$= \sqrt{2 - \frac{2s^{1/2}\|A^{1/2}B^{1/2}\|_*}{(s^{1/2}Tr(A^{1/2})\,Tr(B^{1/2}))}} \tag{6}$$

$$= \sqrt{2 - \frac{2\|A^{1/2}B^{1/2}\|_*}{(Tr(A^{1/2})\,Tr(B^{1/2}))}}$$

$$= d'(A, B)$$

$\square$

## B.3. Signal mismatch distance

In this subsection, we start with motivating the definition of signal mismatch distance by analytically deriving the regression performance mismatch, i.e., the difference between the least squares error of the joint manifold and the average of the least squares error of the individual manifolds. Next, we examine the mathematical properties of the signal mismatch distance and relevant physical interpretations. Finally, we consider a generative model for investigating the properties of the signal mismatch distance through numerical simulations.

### B.3.1. Regression performance mismatch

Recall that we model a manifold $M$ as $\{(x^M(\theta, \psi), \theta)\}$ where $x^M(\theta, \psi)$ is a neural representation parameterized by feature value $\theta$ and in-manifold variability $\psi$. The least squares error of a regressor $w$ is $\langle(w^\top x^M(\theta, \psi) - \theta)^2\rangle_M$ and the first order derivative with respect to $w$ is

$$2\langle x^M(\theta, \psi)(x^M(\theta, \psi))^\top w - \theta \cdot x^M(\theta, \psi)\rangle_M = 2C^M w - 2\bar{x}^M$$

where $\bar{x}^M = \langle[\theta \cdot x^M(\theta, \psi)\rangle_M$ and $C^M = \langle x^M(\theta, \psi)(x^M(\theta, \psi))^\top\rangle_M$ and $\langle\cdot\rangle_M$ refers to taking average over the manifold $M$. Namely, the optimal linear regressor of $M$ (without the biased term, or equivalently, being mean centered) is

$$w^M = (C^M)^{-1}\bar{x}^M$$

and the least squares error is

$$LSE^M = \langle\theta^2\rangle_M - (\bar{x}^M)^\top(C^M)^{-1}(\bar{x}^M).$$

Now, we consider three settings: manifold $A$, manifold $B$, and manifold $A \cup B$. The least squares errors are the following respectively.

$$LSE^A = \langle\theta^2\rangle_A - (\bar{x}^A)^\top(C^A)^{-1}(\bar{x}^A)$$

$$LSE^B = \langle\theta^2\rangle_B - (\bar{x}^B)^\top(C^B)^{-1}(\bar{x}^B)$$

$$LSE^{A\cup B} = \langle\theta^2\rangle_A/2 + \langle\theta^2\rangle_B/2 - (\bar{x}^A + \bar{x}^B)^\top(C^A + C^B)^{-1}(\bar{x}^A + \bar{x}^B)/2$$

where the last equation holds due to $\bar{x}^M = \langle \theta \cdot x(\theta, \psi) \rangle_{A \cup B} = \langle \theta \cdot x(\theta, \psi) \rangle_A / 2 + \langle \theta \cdot x(\theta, \psi) \rangle_B / 2 = \bar{x}^A / 2 + \bar{x}^B / 2$ and $\langle x(\theta, \psi)(x(\theta, \psi))^\top \rangle_{A \cup B} = \langle x(\theta, \psi)(x(\theta, \psi))^\top \rangle_A / 2 + \langle x(\theta, \psi)(x(\theta, \psi))^\top \rangle_B / 2 = C^A / 2 + C^B / 2$. Thus, we arrive Equation 1, Equation 2, and Equation 3 as presented in Section 2.1.

### B.3.2. Properties of the (normalized) signal mismatch distance

As discussed in Section 2.1, the first term (i.e., Equation 1) in the regression performance mismatch (i.e., $LSE^A / 2 + LSE^B / 2 - LSE^{A \cup B}$) motivates the definition of the signal mismatch distance. In the main text, we focus on the normalized signal mismatch distance defined as

$$d_{\text{signal mismatch}}(A, B) = \frac{(\bar{x}^A - \bar{x}^B)^\top (C^A + C^B)^{-1} (\bar{x}^A - \bar{x}^B)}{(\bar{x}^A)^\top (C^A + C^B)^{-1} (\bar{x}^A) + (\bar{x}^B)^\top (C^A + C^B)^{-1} (\bar{x}^B)}.$$

As a remark, in this work we focus on the normalized signal mismatch distance because it nicely serves as a certain "signal-to-correlation ratio" as justified later.

Let us start with enumerating some basic properties of the (normalized) signal mismatch distance.

**Proposition 2.** *For every manifold $A$ and $B$, the following properties hold.*

1. $0 \le d_{\text{signal mismatch}}(A, B) \le 2$.

2. $d_{\text{signal mismatch}}(A, A) = 0$.

3. $d_{\text{signal mismatch}}(A, -A) = 2$.

4. $d_{\text{signal mismatch}}(A, B) = d_{\text{signal mismatch}}(cA, cB)$ *for every* $c \in \mathbb{R}$ *where* $cM = \{(c \cdot x^M(\theta, \psi), \theta)\}$.

5. $d_{\text{signal mismatch}}(A, B) = d_{\text{signal mismatch}}(A(c), B(c))$ *for every* $c \in \mathbb{R}$ *where* $M(c) = \{(x^M(\theta, \psi), c \cdot \theta)\}$.

6. *If manifold $A$ and $B$ are independent, i.e., the subspace spanned by $\{x^A\}$ and the subspace spanned by $\{x^B\}$ only intersect at the origin, then $d_{\text{signal mismatch}}(A, B) = 1$.*

*Proof.*

1. By the fact that $(C^A + C^B)$ is positive semidefinite, we know that the numerator and the two terms in the denominator of the normalized signal mismatch distance are non-negative. Next, expand , we have

$$(\bar{x}^A \pm \bar{x}^B)^\top (C^A + C^B)^{-1} (\bar{x}^A \pm \bar{x}^B)$$
$$= (\bar{x}^A)^\top (C^A + C^B)^{-1} (\bar{x}^A) + (\bar{x}^B)^\top (C^A + C^B)^{-1} (\bar{x}^B) \pm 2(\bar{x}^A)^\top (C^A + C^B)^{-1} (\bar{x}^B).$$

By the non-negativity of the above equation(s), we have

$$|2(\bar{x}^A)^\top (C^A + C^B)^{-1} (\bar{x}^B)| \le (\bar{x}^A)^\top (C^A + C^B)^{-1} (\bar{x}^A) + (\bar{x}^B)^\top (C^A + C^B)^{-1} (\bar{x}^B)$$

Thus, we conclude that $0 \le (\bar{x}^A - \bar{x}^B)^\top (C^A + C^B)^{-1} (\bar{x}^A - \bar{x}^B) \le 2(\bar{x}^A)^\top (C^A + C^B)^{-1} (\bar{x}^A) + 2(\bar{x}^B)^\top (C^A + C^B)^{-1} (\bar{x}^B)$ and hence the normalized signal mismatch distance takes value within $0$ and $2$.

2. As the numerator of the normalized signal mismatch distance is $0$, we conclude that the normalized signal mismatch distance between $A$ and itself is $0$.

3. As the numerator of the normalized signal mismatch distance is $4(\bar{x}^A)^\top (C^A + C^B)^{-1} (\bar{x}^A) + 4(\bar{x}^B)^\top (C^A + C^B)^{-1} (\bar{x}^B$, we conclude that the normalized signal mismatch distance between $A$ and $-A$ is $2$.

4. As (uniformly) scaling the activity patterns by a factor of $c$ will incur a factor of $1/c^2$ in both the numerator and the denominator, the normalized signal mismatch distance won't change.

5. As (uniformly) scaling the stimulus value by a factor of $c$ will incur a factor of $c^2$ in both the numerator and the denominator, the normalized signal mismatch distance won't change.

6. Note that when the subspace spanned by $\{x^A\}$ and the subspace spanned by $\{x^B\}$ only intersect at the origin, we have $(C^A + C^B)^{-1} = (C^A)^{-1} + (C^B)^{-1}$. Thus, we have (i) $(\bar{x}^A - \bar{x}^B)^\top (C^A + C^B)^{-1}(\bar{x}^A - \bar{x}^B) = (\bar{x}^A)^\top (C^A)^{-1}(\bar{x}^A) + (\bar{x}^B)^\top (C^B)^{-1}(\bar{x}^B)$, (ii) $(\bar{x}^A)^\top (C^A + C^B)^{-1}(\bar{x}^A) = (\bar{x}^A)^\top (C^A)^{-1}(\bar{x}^A)$, and (iii) $(\bar{x}^B)^\top (C^A + C^B)^{-1}(\bar{x}^B) = (\bar{x}^B)^\top (C^B)^{-1}(\bar{x}^B)$. Putting everything together, we have that the signal mismatch distance between $A$ and $B$ is 1.

$\square$

The above proposition suggests that the (normalized) signal mismatch distance can be served as a measure ranging from 0 to 2 whereas (i) 0 means the two manifolds are completely aligned; (ii) 1 suggests the two manifolds are independent; (iii) 2 means that the two manifolds are completely misaligned. To sum up, the lower the (normalized) signal mismatch distance, the more correlated the two manifolds in their organization to represent the latent feature.

### B.3.3. Sufficient conditions for the signal mismatch distance capturing regression performance mismatch

Next, we would like to justify that the signal mismatch distance term (i.e., Equation 1) captures the regression performance mismatch when the manifolds satisfy certain sufficient conditions.

**Rotational symmetry.** The simplest way to represent an object is using a low-dimensional ball/sphere. So here we study the signal mismatch distance in this toy setting as a sanity check. In fact, we consider a slightly more general scenario where a manifold enjoys rotational symmetry in the sense that the manifold is rotational symmetric in the subspace of active neurons.

**Proposition 3.** *Suppose both manifold $A$ and manifold $B$ have rotational symmetry and manifolds $A'$, $B'$ have the same collections of activity patterns as manifolds $A$, $B$. If the stimulus representation is also rotational symmetric within each manifold, then*

$$\mathbb{E}_{(A,B)}\left[\left(\frac{LSE^A + LSE^B}{2}\right) - LSE^{A\cup B} - (\bar{x}^A - \bar{x}^B)^\top (C^A + C^B)^{-1}(\bar{x}^A - \bar{x}^B)\right]$$

$$= \mathbb{E}_{(A',B')}\left[\left(\frac{LSE^{A'} + LSE^{B'}}{2}\right) - LSE^{A'\cup B'} - (\bar{x}^{A'} - \bar{x}^{B'})^\top (C^{A'} + C^{B'})^{-1}(\bar{x}^{A'} - \bar{x}^{B'})\right]$$

*where the randomness in the above equation is the direction of encoding the stimulus $\theta$.*

*Proof.* Let $S^A, S^B \subset [n] = \{1, \cdots, n\}$ be the set of active neurons of manifold $A$ and $B$ respectively where $n$ is the number of neurons. Let $n_A = |S^A|$, $n_B = |S^B|$, and $n_{A\cap B} = |S^A \cap S^B|$. In this case, both $C^A$ and $C^B$ are an identity matrix for a subspace in $\mathbb{R}^n$. Let us reindex the neurons so that $S^A = \{1, 2, \ldots, n_A\}$ and $S^B = \{n_A - n_{A\cap B} + 1, \ldots, n_A + n_B - n_{A\cap B}\}$ so that

$$C^A = \begin{pmatrix} I & 0 & 0 & 0 \\ 0 & I & 0 & 0 \\ 0 & 0 & 0 & 0 \\ 0 & 0 & 0 & 0 \end{pmatrix}, \quad C^B = \begin{pmatrix} 0 & 0 & 0 & 0 \\ 0 & I & 0 & 0 \\ 0 & 0 & I & 0 \\ 0 & 0 & 0 & 0 \end{pmatrix}, \quad C^A + C^B = \begin{pmatrix} I & 0 & 0 & 0 \\ 0 & 2I & 0 & 0 \\ 0 & 0 & I & 0 \\ 0 & 0 & 0 & 0 \end{pmatrix}.$$

Notice that if manifold pair $(A, B)$ and $(A', B')$ have the same collections of activity patterns, then $C^A = C^{A'}$ and $C^B = C^{B'}$.

Next, the residual of the difference between regression performance mismatch and the (unnormalized) signal mismatch distance has the following expression:

$$\left(\frac{LSE^A + LSE^B}{2}\right) - LSE^{A\cup B} - (\bar{x}^A - \bar{x}^B)^\top (C^A + C^B)^{-1}(\bar{x}^A - \bar{x}^B)$$

$$= \frac{1}{2}(\bar{x}^A)^\top [(C^A)^{-1} - 2(C^A + C^B)^{-1}](\bar{x}^A) + \frac{1}{2}(\bar{x}^B)^\top [(C^B)^{-1} - 2(C^A + C^B)^{-1}](\bar{x}^B).$$

For convenience, we denote $\hat{C}_1 = (C^A)^{-1} - 2(C^A + C^B)^{-1}$ and $\hat{C}_2 = (C^B)^{-1} - 2(C^A + C^B)^{-1}$. Note that $\hat{C}_1$ and $\hat{C}_2$ are deterministic. Finally, as the stimulus representation within each manifold is also roational symmetric, we have that

$$\mathbb{E}_{(A,B)}\left[(\bar{x}^A)^\top[(C^A)^{-1} - 2(C^A + C^B)^{-1}](\bar{x}^A)\right]$$

$$= \mathbb{E}_{(A,B)}\left[(\bar{x}^A)^\top \hat{C}_1(\bar{x}^A)\right]$$

$$= \mathbb{E}_{(A',B')}\left[(\bar{x}^{A'})^\top \hat{C}_1(\bar{x}^{A'})\right]$$

$$= \mathbb{E}_{(A',B')}\left[(\bar{x}^{A'})^\top[(C^{A'})^{-1} - 2(C^{A'} + C^{B'})^{-1}](\bar{x}^{A'})\right]$$

and similarly for the $B$ and $B'$ term.

We conclude that the two manifold pairs have the same residual of the difference between regression performance mismatch and the (unnormalized) signal mismatch distance in expectation. $\square$

The above proposition can be interpreted as follows. In the simplest non-trivial scenario where the two manifolds are both rotational symmetric, then the residual of the difference between regression performance mismatch and the (unnormalized) signal mismatch distance would not contain any information about the correlation of the latent organization of the two manifolds.

**Matching stimulus marginals.** Before we formally state the result, let us first set up the concept of *am ensemble pair of manifolds*. Mathematically, we say $(\mathbb{A}, \mathbb{B})$ is an ensemble pair of manifolds if it is a distribution of two manifolds $(A, B)$ with fixed collections of activity patterns whereas the associated stimulus value $\theta$ has the same marginal distribution. Physically, this corresponds to invariant representations for two object classes where the underlying latent structures are undetermined. Now, we are able to state the result regarding the connection between signal mismatch distance and the regression performance mismatch in the following proposition.

**Proposition 4.** *Consider two ensemble pair of manifolds $(\mathbb{A}, \mathbb{B})$ and $(\mathbb{A}', \mathbb{B}')$ with the same collections of activity patterns and the same marginal distributions for the stimulus, then we have*

$$\mathbb{E}_{(A,B)\sim(\mathbb{A},\mathbb{B})}\left[\left(\frac{LSE^A + LSE^B}{2}\right) - LSE^{A\cup B} - (\bar{x}^A - \bar{x}^B)^\top(C^A + C^B)^{-1}(\bar{x}^A - \bar{x}^B)\right]$$

$$= \mathbb{E}_{(A',B')\sim(\mathbb{A}',\mathbb{B}')}\left[\left(\frac{LSE^{A'} + LSE^{B'}}{2}\right) - LSE^{A'\cup B'} - (\bar{x}^{A'} - \bar{x}^{B'})^\top(C^{A'} + C^{B'})^{-1}(\bar{x}^{A'} - \bar{x}^{B'})\right].$$

*Proof.* First, the residual of the difference between regression performance mismatch and the (unnormalized) signal mismatch distance has the following expression:

$$\left(\frac{LSE^A + LSE^B}{2}\right) - LSE^{A\cup B} - (\bar{x}^A - \bar{x}^B)^\top(C^A + C^B)^{-1}(\bar{x}^A - \bar{x}^B)$$

$$= \frac{1}{2}(\bar{x}^A)^\top[(C^A)^{-1} - 2(C^A + C^B)^{-1}](\bar{x}^A) + \frac{1}{2}(\bar{x}^B)^\top[(C^B)^{-1} - 2(C^A + C^B)^{-1}](\bar{x}^B).$$

Next, as the two ensemble pairs $(\mathbb{A}, \mathbb{B})$ and $(\mathbb{A}', \mathbb{B}')$ share the same collections of fixed activity patterns, we have $C^A = C^{A'}$ and $C^B = C^{B'}$. Moreover, as the two ensemble pairs also share the same marginal distribution (of the stimulus $\theta$), we also have $\mathbb{E}[f(\bar{x}^A, \bar{x}^B)] = \mathbb{E}[f(\bar{x}^{A'}, \bar{x}^{B'})]$ for any function $f$. Thus, we have

$$\frac{1}{2}(\bar{x}^A)^\top[(C^A)^{-1} - 2(C^A + C^B)^{-1}](\bar{x}^A) + \frac{1}{2}(\bar{x}^B)^\top[(C^B)^{-1} - 2(C^A + C^B)^{-1}](\bar{x}^B)$$

$$= \frac{1}{2}(\bar{x}^{A'})^\top[(C^{A'})^{-1} - 2(C^{A'} + C^{B'})^{-1}](\bar{x}^{A'}) + \frac{1}{2}(\bar{x}^{B'})^\top[(C^{B'})^{-1} - 2(C^{A'} + C^{B'})^{-1}](\bar{x}^{B'}).$$

We conclude that the two ensemble pairs have the same residual of the difference between regression performance mismatch and the (unnormalized) signal mismatch distance. $\square$

The above proposition can be interpreted as follows. An ensemble pair of manifolds corresponds to a (mathematical) model for how a neural network represent two potentially correlated concepts with a shared latent feature. The two assumptions (fixed activity patterns and matching marginals) correspond to the empirical observations/representations one can record from the neural network. Note that there could be many ensemble pairs of manifolds having the same collections of activity patterns and marginal distributions, while some of them might have correlated latent structure (across manifolds), and some might not. The above proposition essentially says that the residual of the difference between regression performance mismatch and the (unnormalized) signal mismatch distance would not contain information about the correlation between the two manifolds.

### B.3.4. A generative model and numerical simulations

In this subsection we define and investigate a synthetic model and its variants to understand the properties of the signal correlation mismatch distance.

Let $n$ be the number of neurons and let $m$ be the number of points in each manifold. Also, let $\tau \in [0, 1]$ be a correlation parameter and let $\epsilon \geq 0$ be a noise parameter. We generate the sample points and features of each manifold via the following process.

1. Randomly sample $m$ independent points from the $n$-dimensional Gaussian distribution with mean $0$ and covariance $I$. Collect these points into manifold $A$.
2. Generate the points in manifold $B$ using the same procedure.
3. Randomly sample an unit vector $w_A$ in $\mathbb{R}^n$.
4. Randomly sample another unit vector $w$ in $\mathbb{R}^n$ and let $w_B$ be the unit vector in the direction of $(1 - \tau)w_A + \tau w$ where $\tau \in [0, 1]$ is a parameter representing the correlation between the two manifolds.
5. For each point $x$ in the manifold $A$ (resp. $B$), associate it with feature value $\theta = w_A^\top x + \epsilon \xi$ (resp. $\theta = w_B^\top x + \epsilon \xi$) where $\xi$ is a noise term sampled from the standard Gaussian distribution.

We measure the (normalized) signal correlation mismatch distance of the above generative models and plot the results in Figure 8.

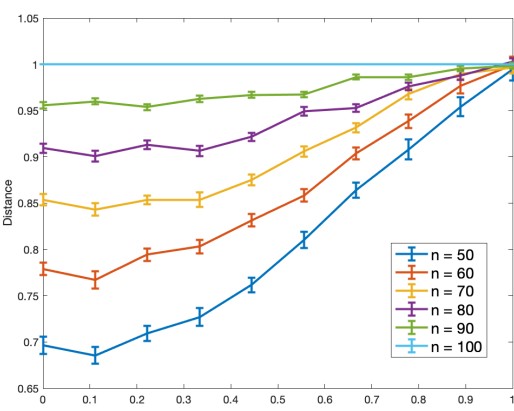

Figure 8: Simulations with $m = 50$, $n = 50, 60, 70, 80, 90, 100$, $\epsilon = 0.05$, and $\tau = 0, 0.1, 0.2, \ldots, 1$.

Intuitively, the smaller the $\tau$ is, the more aligned the two manifolds are. Namely, we expect to see that the signal correlation mismatch distance grows as $\tau$ goes from 0 to 1 and indeed that's what happen in the numerical simulations as shown in Figure 8. Next, the increase in distance with respect to the growth of the number of neurons is a result of insufficient number of samples, which we will address in the following.

**Manifolds coming from lower dimensional subspaces** The previous simulation suggests that when the ambient dimension $n$ (i.e., the number of neurons) is much greater than the number of samples $m$, then we would not have enough points to estimate the distance correctly because the manifold dimension is intrinsically $n$. Meanwhile, in real data, we would expect the manifolds having a low intrinsic dimension and/or even overlap with each other in some subspaces. So here we consider a variant of the above generative model to reconcile these issues. Let $d_M$ be a parameter for the dimension of each manifold and let $d_{over}$ be a parameter for the dimension of the overlapped subspace of the two manifolds.

1. Randomly sample $m$ independent points from the $d_M$-dimensional Gaussian distribution with mean $0$ and covariance $I$ in the $n$-dimensional ambient space. Collect these points into manifold $A$.

2. Generate the points in manifold $B$ using the same procedure while randomly picking a $d_{over}$-dimensional subspace from manifold $A$ into the subspace of manifold $B$.

3. Randomly sample an unit vector $w_A$ in $\mathbb{R}^n$.

4. Randomly sample another unit vector $w$ in $\mathbb{R}^n$ and let $w_B$ be the unit vector in the direction of $(1-\tau)w_A + \tau w$ where $\tau \in [0,1]$ is a parameter representing the correlation between the two manifolds.

5. For each point $x$ in the manifold $A$ (resp. $B$), associate it with feature value $\theta = w_A^\top x + \epsilon\xi$ (resp. $\theta = w_B^\top x + \epsilon\xi$) where $\xi$ is a noise term sampled from the standard Gaussian distribution.

We numerically simulate different choices of overlapped dimension and plot the signal correlation mismatch distance as well as the regression loss in Figure 9.

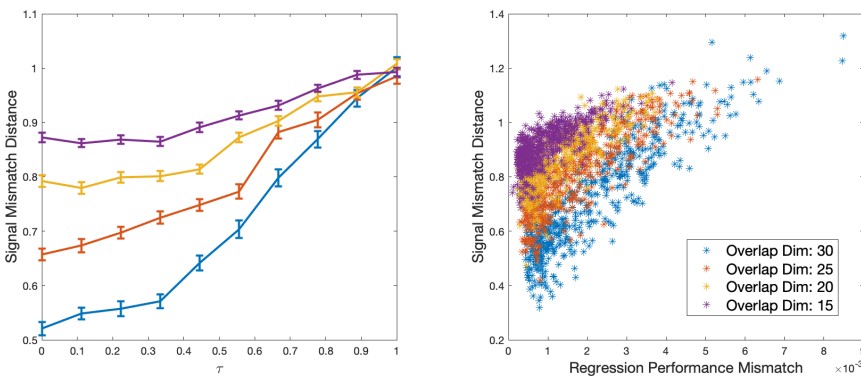

Figure 9: Simulations with $m = 50$, $n = 100$, $\epsilon = 0.05$, $\tau = 0, 0.1, 0.2, \ldots, 1$, $d_M = 30$, $d_{over} = 15, 20, 25, 30$.

Intuitively, we expect the distance to be smaller when the two manifolds overlap with each other more. And this is confirmed in Figure 9(a). Moreover, we postulated earlier that the signal correlation mismatch distance will positively correlates with the regression loss (i.e., the loss of optimal regressor for both manifolds). Indeed we confirm this connection in the generative model as shown in Figure 9(b).

To sum up, we design a generative model of correlated manifolds with an 1D latent structure and test the (normalized) signal mismatch distance. Both Figure 8 and Figure 9 suggest that the (normalized) signal mismatch distance captures the latent correlation across the two manifolds as well as explains the regression performance mismatch as expected.

## B.4. Cross-Regression Performance Additional Details

As mentioned in Section 3.4, the cross-regression performance is R2 achieved from training a linear regression model on one latent variation manifold around an exemplar and evaluating on another. More formally, the cross-regression performance from manifold A to manifold B is obtained via the following procedure:

- Train a ridge regression model (linear regression with L2-normalization) on a manifold A.
- Evaluate the $R^2$ of the ridge regression model on manifold B.

The resulting $R^2$ value is the cross-regression performance from A to B.

## B.5. Model details

| Model | ImageNet Top-1 Accuracy (%) | Source |
|---|---|---|
| Supervised | 76.15 | Torchvision |
| Supervised-RE | 78.47 | PyTorch Image Models [46] |
| Supervised-RARE | 78.81 | PyTorch Image Models |
| Barlow Twins | 73.5 | Facebook Research (PyTorch Hub) |
| SwAV | 75.3 | Facebook Research (PyTorch Hub) |
| VICReg | 73.2 | Facebook Research (PyTorch Hub) |
| DeepClusterV2 | 75.18 | Facebook Research (VISSL) |
| SimCLR | 68.8 | Facebook Research (VISSL) |
| PIRL | 64.29 | Facebook Research (VISSL) |
| BYOL | 74.6 | DeepMind & PyTorch Conversion [2] |

All models used a ResNet-50 architecture.

---

[2]https://github.com/ajtejankar/byol-convert

## B.6. Deaggregated results

### B.6.1. MCT Results

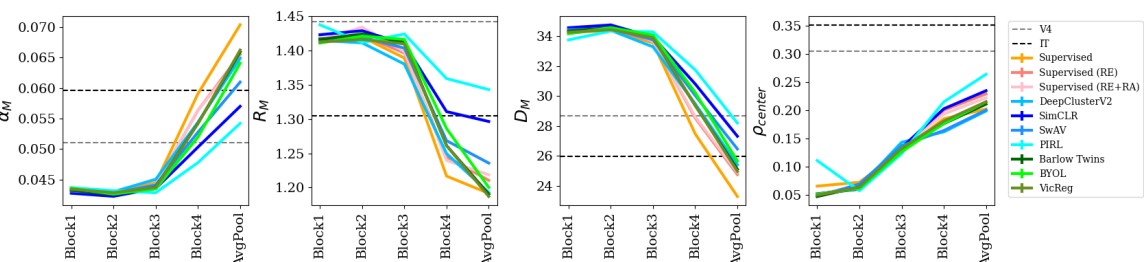

Figure 10: As in Figure 4 we extracted the hidden representations of each image from the output of the four ResNet blocks in each model, and the terminal average pool layer and computed the manifold capacity, radius, dimension, and center correlations across the layers of the DNNs. We also performed MCT analysis on macaque brain recording (dashed lines).

### B.6.2. MAA Results

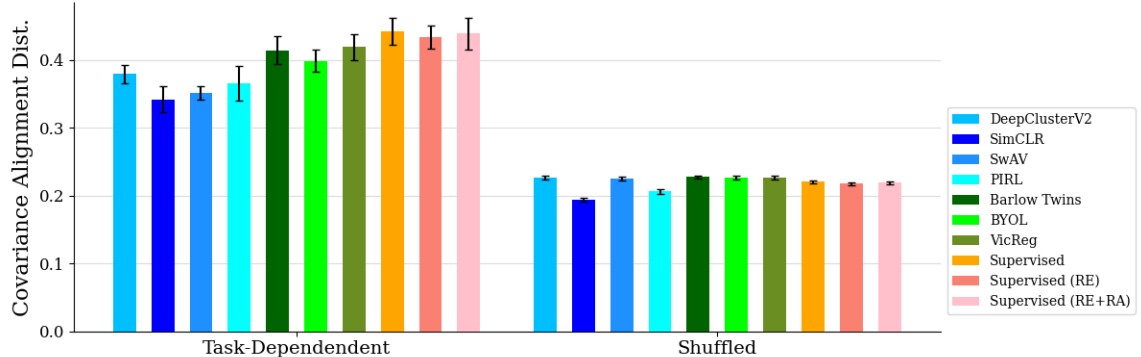

Figure 11: As in Figure 5, we averaged the trace-normalized Bures covariance distance between the 8 super category manifolds in our dataset to quantify how aligned the object manifolds were in each model. We also repeat the experiment, but with shuffled manifolds (random partitioning).

### B.6.3. Neural Comparison (Relative Analysis) Results

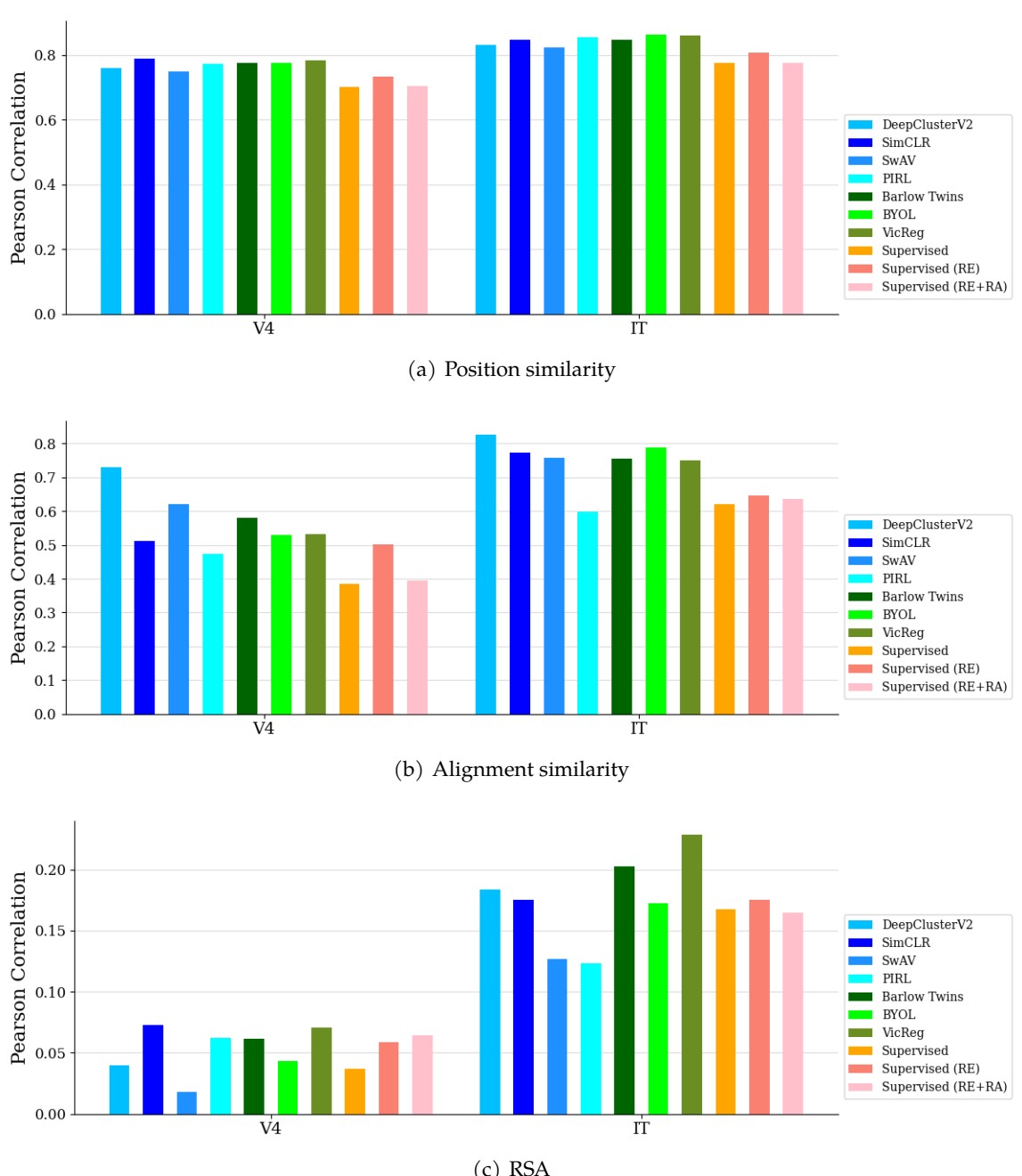

(a) Position similarity

(b) Alignment similarity

(c) RSA

Figure 12: Deaggragated results of Figure 6, showing similarity to macaque neural data. RSA fails to provide consistent trends when comparing supervised to unsupervised models.

## B.7. Code and Compute

The code used for new experiments will be released to the public. The code used for Manifold Capacity Theory (MCT) analysis was obtained from: https://github.com/schung039/neural_manifolds_replicaMFT

These experiments were run on a computing cluster with about 1000 multicore nodes with up to 1TB of memory each. GPUs were not used in these experiments.

