# OpenReview forum: "Probing Biological and Artificial Neural Networks with Task-dependent Neural Manifolds"
_CPAL.cc/2024/Conference — CPAL 2024 (Proceedings Track) Oral_

### Meta-Review · Area_Chair_7xU4 · 2023-11-15

**Recommendation:** Accept (Oral)
**Confidence:** 5

**Metareview:**

(this is an invited paper, so only one meta-review is provided)

Machine learning, and deep learning in particular, have drawn significant
inspiration from neuroscience over the years, with far-reaching impacts.
This submission, along with an emerging body of exciting research, demonstrates
that this statement can also be turned around: deep learning can
provide insights and inspirations for research into the nature of representation
learning in both artificial and biological neural networks.
To this end, the authors conduct a detailed study into how representations in
artificial (ResNet backbone) and biological (in vivo micro-electrode recordings
of V4 and IT in macaques) neural networks differ from one another as a function
of task-specific aspects, such as supervised versus unsupervised training as
well as object-specific or class-specific conditioning.
They base this study on computational metrics related to the geometry of
representations -- specifically, metrics from manifold capacity theory (MCT),
advanced by Chung et al. (2018) with tools from statistical physics, are used to
quantify how well representations of different classes can be separated from one
another, and metrics from manifold alignment analysis (MAA) are used/proposed to
quantify how well representations of different classes align with one another
(e.g., in the sense of disentangled representation learning). The work
contributes to MAA by proposing a new metric, the signal mismatch distance, with
detailed calculations and verifications in simple models given in the
supplemental.

Their experiments obtain artificial and biological neural representations via
stimulation using visual inputs consisting of one of a set of controlled 3D
objects (6D pose) embedded into a background, for the different pretrained deep
networks and macaques.
Within this context, the authors present three classes of findings.
The first is that unsupervised learning methods for deep networks (BYOL, SimCLR,
many more)
learn representations with systematically lower MCT scores than supervised
learning methods, suggesting less task-specialized (i.e., classification)
behavior of the learned representations; these findings correlate with the
observations for V4 (earlier in the visual cortex) and IT. The second is that
unsupervised deep network representations display greater MAA scores than
supervised representations, and that this extends to unsupervised deep
representations showing greater alignment similarity to neural representations
from the macaque visual cortex. Interestingly, these experiments demonstrate
that a standard neural data analysis technique fails to distinguish these
similarities -- it is necessary to leverage task-specific information in the
comparison between artificial and biological representations, as in the MAA
metrics. The third is that when controlling the visual stimulus to vary along a
single degree of freedom (fixed object, varying pose), representational
alignment (measured through MAA metrics) typically correlates positively with
various indicators of downstream performance.

Taken together, these results suggest that metrics based on data geometry
provide a useful avenue to analyze representations in biological and artificial
neural networks, as well as interesting avenues for future work, especially
surrounding better understanding of the novel MAA metrics.
These results provide a fresh perspective on representation learning, amidst
other emerging theories for representation learning based on, for example,
neural collapse and compression, and will make a valuable addition to CPAL.

For the camera-ready, the authors may wish to correct a few minor points:
removing lines 465-466; changing $\\| AB \\|$
to $\\| A^{1/2} B^{1/2} \\|$ (both nuclear norms) in eqn. (4).

---

### Decision · Program_Chairs · 2023-11-20

Accept (Oral)